# Basic reproduction number varies markedly between closely related pandemic *Escherichia coli* clones

Fanni Ojala [1,11], Henri Pesonen [2,3,11], Rebecca A. Gladstone [4], Tommi Mäklin [5], Gerry Tonkin-Hill [4,6,7,8], Pekka Marttinen [1,12] & Jukka Corander [4,5,9,10,12] ✉

Extra-intestinal pathogenic *Escherichia coli* ubiquitously colonize the human gut and represent clinically the most significant bacterial species causing urinary tract infections and bacteremia. During the last two decades clades of the ST131 lineage have spread globally, but it remains unknown how their transmission dynamics compare to the basic reproduction numbers ($R_0$) for viral pandemics. We develop a compartmental model for asymptomatic gut colonization and onward transmission coupled with an epidemiological observation model and fit it on the major ST131 clades. Our results indicate that the ST131-A transmission potential ($R_0 = 1.47$) can be comparable to pandemic influenza viruses, while the significantly lower transmissibility of ST131-C1 ($R_0 = 1.18$) and ST131-C2 ($R_0 = 1.13$) suggests that their dissemination has been aided by antibiotic selection pressure and healthcare facilities. Our results provide an advance in understanding the relative transmissibility of these opportunistic pathogens.

As evidenced by extensive research on emerging pathogens over the last couple of decades, such as Ebola, H1N1 influenza, SARS, MERS and SARS-CoV-2, epidemiological modeling of their transmission rates provides a key piece of information that serves many purposes, including the forecasting of future spread and design of interventions. For a critical overview of different methods for inferring the basic reproduction number $R_0$, and their reliability, see[1–5]. Different compartmental models can be used to model the spread of an infectious disease in a population, out of which the most basic one is the Susceptible-Infected-Recovered (SIR) model (for details see the above references). For emerging pathogenic viruses the incubation period

tends to be short, typically a matter of days, which translates to an opportunity to trace and count cases by reporting symptomatic individuals in a population on a given day or week of an ongoing epidemic. These data represent the fundamental information needed in modeling the spread of the pathogen in such a setting, irrespective of whether deterministic or stochastic models are considered. For bacteria the situation is much more complicated and consequently $R_0$ is rarely inferred. The notable exceptions where $R_0$ has been estimated are highly virulent bacteria causing local outbreaks, such as *Neisseria meningitidis*[6] and TB[7], or bacterial strains in more closed ecological settings such as animal production facilities where their spread can be

[1]Department of Computer Science, Aalto University, Konemiehentie 2, Espoo, Finland. [2]Oslo Center for Biostatistics and Epidemiology, Oslo University Hospital, Sognsvansveien 9, Oslo, Norway. [3]Faculty of Information Technology and Communication Sciences, Tampere University, Korkeakoulunkatu 1, Tampere, Finland. [4]Department of Biostatistics, University of Oslo, Sognsvansveien 9, Oslo, Norway. [5]Helsinki Institute for Information Technology HIIT, Department of Mathematics and Statistics, University of Helsinki, Yliopistonkatu 3, Helsinki, Finland. [6]Peter MacCallum Cancer Centre, 305 Grattan Street, Melbourne, Australia. [7]Sir Peter MacCallum Department of Oncology, The University of Melbourne, Grattan Street, Melbourne, Australia. [8]Department of Microbiology and Immunology, The University of Melbourne at the Peter Doherty Institute for Infection and Immunity, 792 Elizabeth Street, Melbourne, Australia. [9]Parasites and Microbes, Wellcome Sanger Institute, Wellcome Genome Campus, Hinxton, Cambridgeshire, United Kingdom. [10]Department of Genetics, University of Cambridge, Downing Street, Cambridge, United Kingdom. [11]These authors contributed equally: Fanni Ojala, Henri Pesonen. [12]These authors jointly supervised this work: Pekka Marttinen, Jukka Corander. ✉e-mail: jukka.corander@medisin.uio.no

traced for example using seroconversion data[8]. Outside particularly vulnerable populations, e.g., hospital patients, common opportunistic pathogenic bacteria are frequently carried asymptomatically, potentially for months or years, and reports of infections caused by them can occur with a considerable time lag after the colonization event. Therefore determining the length of time between the acquisition of the pathogen to an infection or transmission is not feasible even in the most advanced and systematic public health surveillance systems.

Bacteria causing hidden pandemics can be broadly categorized into two distinct types, those which are strongly associated with nosocomial infections and are mainly spread through healthcare networks and those which predominantly spread in the community as part of the commensal microbiota of the gut, skin or the nasopharynx[9,10]. Extra-intestinal pathogenic *Escherichia coli* (ExPEC) represent the latter category and are among the clinically most significant causes of human bacterial infections, such as bacteremia and urinary tract infections[11]. During the last decades several so-called pandemic clones of ExPEC have emerged and these are typically characterized by both high levels of virulence and antibiotic resistance[12,13]. For three particularly famous clades belonging to the pandemic ST131 clone, ST131-A, ST131-C1 and ST131-C2, previous work has inferred the timing and rates of their expansions using a Bayesian phylodynamic approach which provides an estimate of the effective population size over time[14]. Using systematic genomic surveillance of *E. coli* bloodstream infections (BSIs) in Norway over 2002-2017, it was shown that these clones likely expanded in parallel in the host population around early 2000s but also that ST131-A appeared to have markedly higher dissemination success than ST131-C1 and ST131-C2. In contrast, based on phylodynamics the clade ST131-B had already expanded earlier in the 1990s, and other well-known pandemic ExPEC clones, such as ST69, ST73 and ST95, were also more recently demonstrated to have established themselves earlier such that no signal of sufficiently recent expansion was present in the Norwegian surveillance data[15]. In summary, these results put forward the specific ST131 clades as attractive targets for epidemiological modeling to try to quantify their transmission efficiencies relative to each other and in a broader context in comparison with other microbes exhibiting pandemic spread.

Since systematic surveillance of asymptomatic colonization rates of *E. coli* clones has been missing, it has not been possible to turn the disease surveillance data into similar epidemiological insight as captured by $R_0$. Recently, an advance in this direction was made by Mäklin et al. who inferred strain-level colonization competition dynamics from shotgun metagenomics data for a large neonatal birth cohort in the UK and for several bacterial pathogens, including *E. coli*[16]. By combining these data with temporally matched genomic *E. coli* disease surveillance from the UK, it was possible to infer the relative success of colonization vs infection for the ST131 clades[17] (see also[18]). Further, this information can be merged with the comprehensive population level bloodstream infection statistics available for Norway and the annual incidences of ST131-A, ST131-C1 and ST131-C2 in Norway based on the published genomic surveillance results. Here, we harness the opportunity presented by the combination of these data, and develop a compartmental model for the asymptomatic colonization and transmission of these three *E. coli* clades of ST131 in Norway and couple this with a stochastic observation model for bloodstream infections, to estimate $R_0$ using simulation-based inference. Comparison of their values provides useful epidemiological insight and allows relating them more directly to other pandemic pathogens in terms of the rate of their spread.

## Results

In order to link the asymptomatic spread of *E. coli* in the host population with the reported bloodstream infections from the national surveillance in Norway, we developed a hybrid model with both a deterministic and a stochastic component. The model structure is illustrated in Fig. 1. Our model consists of an unobserved compartmental SIR-like colonization model for the host population at the national level and an observation model which links the colonization with observed annual bloodstream infection incidence for clades A, C1 and C2. The observation model further uses clade-specific estimated odds ratios of causing a bloodstream infection. These odds ratios compare the frequency of asymptomatic colonization to the frequency of bloodstream infection in a temporally and geographically matched host population (UK) from which data based on representative sampling of hosts were available[16,17]. Full details of the model are provided in the Methods section.

Posterior distributions of the basic reproduction number $R_0$ parameters are visualized for the three *E. coli* ST131 clades in Fig. 2. Figure 3 correspondingly shows the marginal and joint posterior distribution of the basic reproduction number $R_0$ and the net transmission rate. Posterior means and 95% posterior credible intervals for the three parameters are presented in Table 1. The posterior distributions indicate a markedly higher $R_0$ for clade A, translating to markedly better transmission efficiency in the underlying population.

The model fit to data and the 50/95% credible intervals for predicted BSI counts are visualized in Fig. 4, showing that the hybrid model structure captures the major differences between the two clades. Note that by definition the SIR-based model includes a temporal peak in colonizations, and consequently there is a corresponding decline in colonization and the predicted BSI counts beyond the peak as seen from the figure.

Figure 5 visualizes the distribution of predicted weekly colonization counts in a population of 1,000,000 individuals, with parameters $\tau$, $R_0$ and $\Delta$ integrated out by sampling from the posterior. These results suggest approximately ten-fold peak colonization success for ST131-A compared with the other two clades. Of note, since the SIR model fit is intended for the calendar time window where the exponential growth of colonization success took place, these predictions should not be extrapolated into the future where the population prevalence of each clade has reached an equilibrium.

To assess sensitivity of the obtained results under the chosen modeling framework and uncertainty about key assumptions, we also fitted the model with alternative values of the invasiveness parameter based on their confidence intervals from a previous study[17] and different priors for the delay parameter (Supplementary Information). The sensitivity analysis shows that the interpretation of a significantly higher transmission potential of ST131-A compared with ST131-C1/ST131-C2 remains robust as a function of varying these assumptions (Supplementary Information). Further, we included a formal comparison of the SIR-based model with an alternative SIS-based model lacking the recovered compartment. Generally the fit of the latter model to the BSI data was much poorer and the per clade acceptance rates of samples from the posterior were at least an order of magnitude lower for the SIS- vs. SIR-based model, quantitatively demonstrating its inferiority to capture the observed expansion patterns (Supplementary Information).

## Discussion

While $R_0$ represents a key epidemiological characteristic that is typically rapidly inferred for emerging pathogenic viruses, its quantification has remained far more elusive for pathogenic bacteria causing opportunistic infections. The primary reasons behind this difficulty are the lack of systematic high-resolution surveillance of asymptomatic colonization rates for almost any bacterial species, the temporal lag with which infection cases tend to occur, and the need to have a system in place for whole-genome sequencing of infection isolates from sufficiently representative surveillance. Here we were able to fill this knowledge gap by combining systematic population genomics based surveillance of *E. coli* bloodstream infection isolates in both Norway

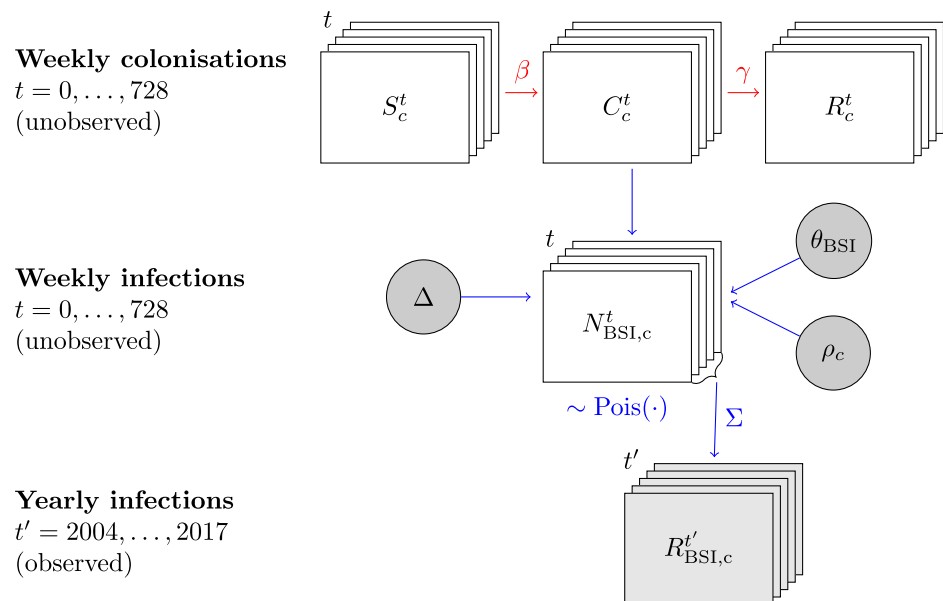

**Fig. 1 | Overview of the hybrid deterministic-stochastic model structure.** The first row shows the latent population colonization model, implemented by simulating outcomes from a compartmental SIR-like model. The compartments $S_c^t$, $C_c^t$ and $R_c^t$ are the numbers of susceptible, colonized, and removed individuals in week $t$ for clade $c$. The transmission coefficient $\beta$ and the recovery rate $\gamma$ describe the flow between the compartments and in our implementation, the model is parametrized in terms of the basic reproduction number and net transmission rate, see[31]. The observation model is visualized below the colonization model, comprising the

weekly simulated bloodstream infection (BSI) counts $N_{BSI,c}^t$ with the aggregated yearly BSI infection counts $R_{BSI,c}^{t'}$, for which observed values are available from the national surveillance system. In the observation part, parameter $\rho_c$ is the estimated relative invasiveness of clade $c$, $\theta_{BSI}$ is the fraction of the population experiencing an *E. coli* BSI caused by any clade and $\Delta$ specifies the number of years simulated colonization processes precede the observed data, essentially representing an expected time lag between colonization and infection. A Poisson observation model is used to invoke stochastic variation in the simulated BSI case counts.

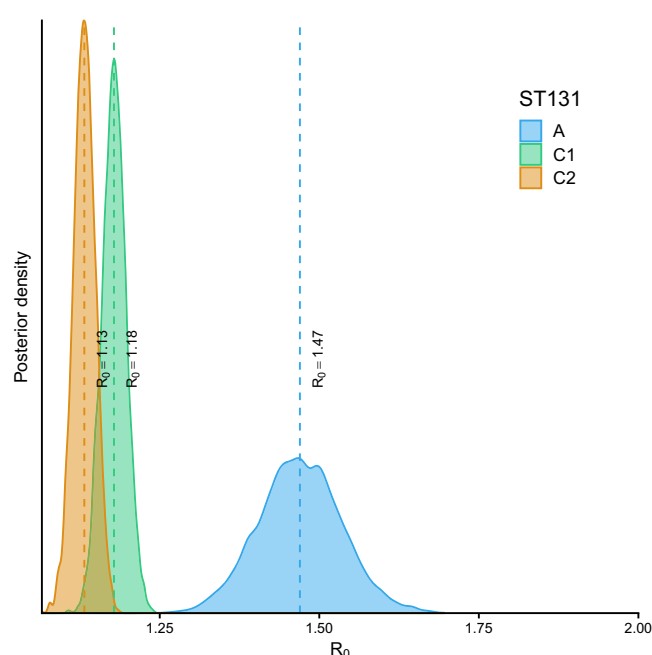

**Fig. 2 |** Posterior distributions of the basic reproduction number $R_0$ for ST131-A, ST131-C1 and ST131-C2, with median point estimate of $R_0$ highlighted by a dashed line.

and the UK, with demographics and annual national-level mandatory reporting statistics of these infections in Norway. Furthermore, we had the opportunity to use recently estimated odds ratios for each considered clade of *E. coli*[17], which compare the frequency of asymptomatic colonization to the post-expansion frequency of bloodstream

infections in a temporally matched host population in the UK. Further comparative analysis using published asymptomatic colonization data from adults in France was also conducted to ensure the UK data were generally sufficiently representative in terms of geography and colonization age (Methods). Changes in clinical practice during the observation window could also introduce a bias in these estimates. However, the infection surveillance systems from which the original study isolates were acquired in UK and Norway operate under similar principles and aim to be representative on a national level, which should reduce the risk of systematic biases influencing comparisons.

By using all the above-mentioned data sources, we built a hybrid deterministic-stochastic model that first simulates the asymptomatic spread of a novel bacterial clone in a population and then feeds this information into an observation model that generates infection cases at the desired temporal resolution. Unlike the basic epidemiological models, our approach results in an intractable likelihood, calling for simulator-based inference methods, such as ABC[19], to calibrate the model parameters. Recently, Pesonen et al. demonstrated the extended possibilities offered by ABC inference for simulator-based modelling of $R_0$ when the transmission process exhibits latent heterogeneity[20], using a model derived for the Ebola virus by Britton and Tomba[21]. Despite our focus on *E. coli*, we believe that the current study can serve as a proof-of-concept for modelling expansions of other bacteria with pathogenic potential and thus extend the reach of infectious disease epidemiology to further applications. However, a limitation of our approach is that it requires either asymptotic colonization or infection surveillance data that are sampled in a representative manner during the expansion of a bacterial lineage, in contrast to phylogenetic methods, which can also infer past population expansions from samples obtained once a pathogen has reached endemicity.

Compartmental models for transmission studies are generally built from assumptions that might not hold in reality, such as homogeneous mixing of host population (in the case of *E. coli*

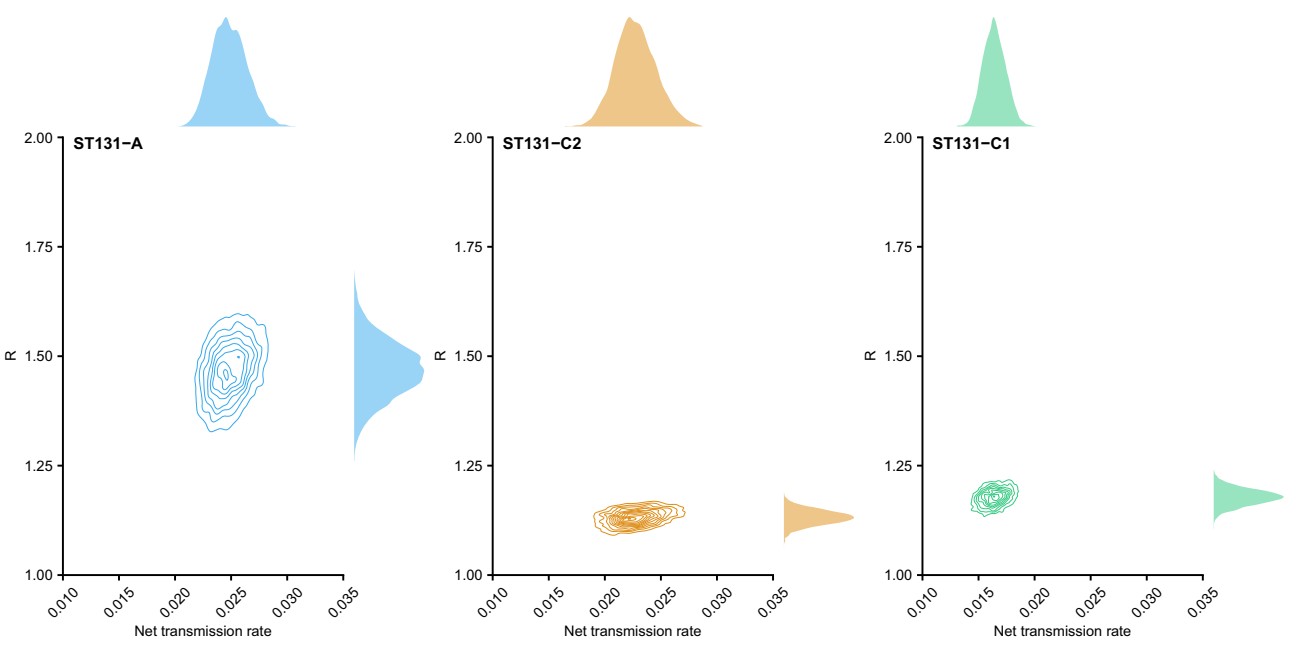

**Fig. 3 |** Joint and marginal posterior distributions of the basic reproduction number $R_0$ and the net transmission rate parameters for ST131-A, ST131-C1 and ST131-C2.

**Table 1 | Means and 95% credible intervals for the basic reproduction number $R_0$ and the net transmission rate**

| | ST131-A | | ST131-C2 | | ST131-C1 | |
|---|---|---|---|---|---|---|
| Parameter | Mean | 95% CI | Mean | 95% CI | Mean | 95% CI |
| $R_0$ | 1.47 | (1.30, 1.60) | 1.13 | (1.08, 1.20) | 1.18 | (1.12, 1.20) |
| Net transmission rate | 0.025 | (0.021, 0.028) | 0.023 | (0.018, 0.027) | 0.016 | (0.014, 0.018) |

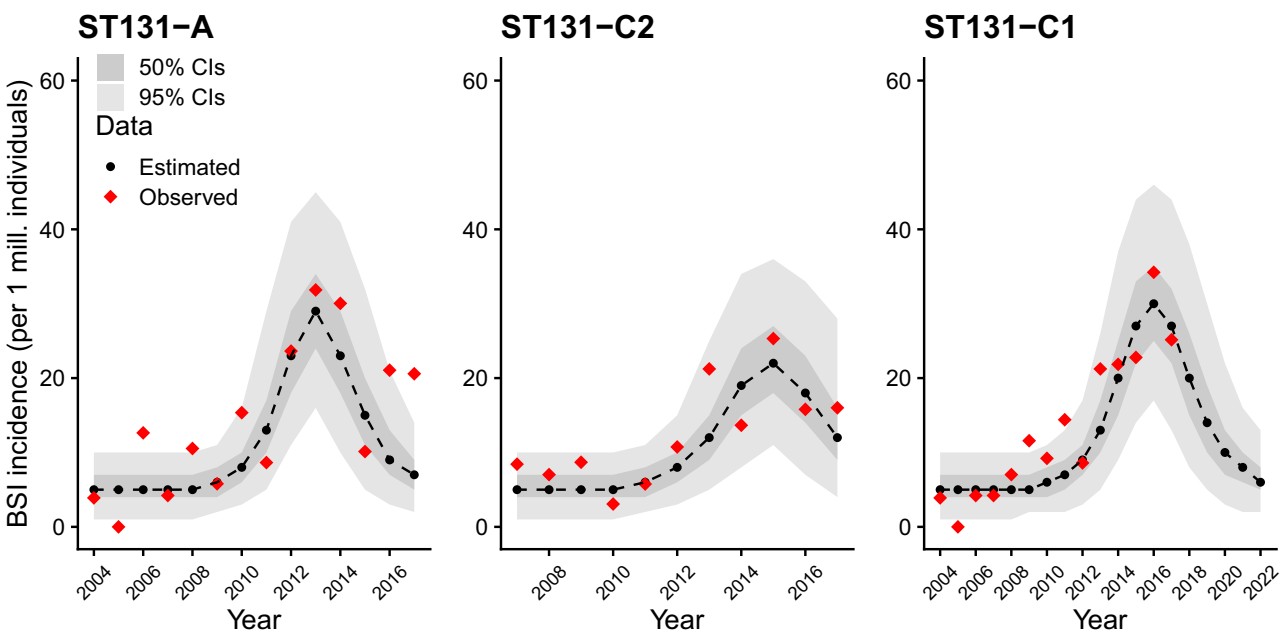

**Fig. 4 | Predictive fit of the model to the observed data.** The red diamonds represent observed bloodstream infection (BSI) counts and the black dots connected by a dashed line represent the median fit for each year. Credible intervals (CI) are shown as two shaded regions corresponding to 50% and 95% credibility levels. Source data are provided as a Source Data file.

transmission events are in reality expected to more likely within households given the oral-fecal route of transmission), mass action incidence to determine rate of new infections, effective contacts (contact always leads to successful transmission) and unchanging demographics on the relevant timescale[22]. Despite these simplifying assumptions, even basic compartmental models have been widely popular and successfully used for a large number of studies in infectious disease epidemiology.

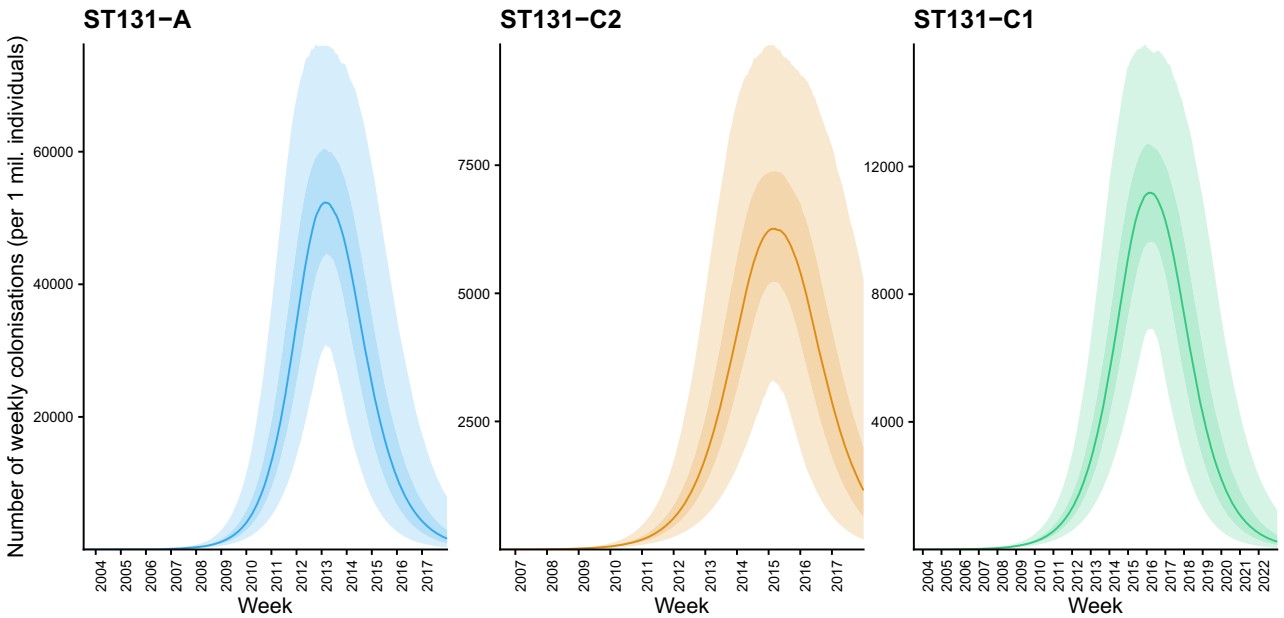

**Fig. 5 | Estimated median weekly colonization counts in a population of 1,000,000 individuals for ST131-A, ST131-C1 and ST131-C2.** The darker and lighter shaded areas around the median line correspond to the 50% and 95% credible intervals, respectively.

We considered an alternative compartmental model for the bacterial colonization using the SIS-based modeling assumption[22]. Here, re-colonization of the same individual is possible and there is no resistance acquired against colonization, leading to exclusion of the recovered compartment. However, quantitative comparison of the models showed that the SIS-based model architecture led to a considerably poorer data fit (Supplementary Information).

We did not include any explicit age structure in our model, although its use could be motivated by the higher incidence of BSI in the elderly population. Nevertheless, it is justifiable to consider that the expanding clades are equally capable of colonizing hosts irrespective of their age, translating to model fits where the relative difference between the clades is robustly quantified. To better capture the variability in incidence at the beginning of the clade expansion, a stochastic branching process based on contact networks could be used in future work in addition to the deterministic compartmental model[22]. Despite the deterministic nature of the SIR-like model for colonization our results show a reasonable degree of fit both in the earliest phase of expansion and during the peaking incidence (Fig. 4). It is also worth noting that our model treats the clades independently, thus excluding potential interactions such as ecological competition, cross-immunity and within-host dynamics. Expanding the modeling assumptions by including some of these biological processes in future work could provide further refined insight into the relative transmission capabilities of the different clades.

Our results indicate a substantially higher $R_0$ for clade A compared with clades C1, C2 (95% credible intervals are non-overlapping), which translates to a markedly higher transmission potential of A that is in principle comparable to some pandemic influenza strains such as the 2009 H1N1[23], albeit caution is advocated when comparing so different pathogens whose transmission is also not based on the same route (oral-fecal vs. airborne route). Our main finding is thus well aligned with the earlier Bayesian phylodynamic inference based finding of significantly higher increase in the effective population size of A clade during the mid-2000s[14]. The significantly lower estimated transmissibility of ST131-C1 and ST131-C2 suggests that its global dissemination has been substantially aided by antibiotic selection pressure and that it may be more effectively transmitted through health care facilities instead of primarily community driven transmission. For

further discussion on the drivers of clade C2 equilibrium prevalence in a population, see[17]. Importantly, clades of the ST131 clone have diverged quite recently and are still closely related in terms of their core genomes[14,24,25]. It is thus remarkable how different they are in terms of their estimated ability to colonize new hosts, which highlights the need for further research on whether this difference can be explained by subtle genetic diversity in their core genomes, such as in metabolic capacity, or by variable accessory genome content (or both). These topics have been a subject of intensive recent research efforts in *E. coli*[24–30], but thus far no clear conclusions have been reached concerning the main genetic drivers underlying phenotypic variation at this level of relatedness. Improved in vitro and in vivo experiments combined with high-resolution comparative population genomics and further modelling could thus offer a promising strategy towards resolving the likely polygenic basis of the estimated difference in transmission efficiency.

## Methods
### Model and inference overview
Our modeling framework, illustrated in Fig. 1, consists of two components: a colonization model, represented by the standard compartmental SIR (susceptible-infected-recovered) model, and an observation model to connect the simulated colonizations to the incidence of bloodstream infections (BSI). The specific *E. coli* clade whose dynamics is being considered is denoted by $c$. In the SIR-like model, the $I$ compartment is replaced by $C_c$ (colonization) that represents the number of individuals colonized by a clade $c$. The parameters of interest in the colonization model are the transmission coefficient $\beta$ and the recovery rate $\gamma$. The reproduction number $R_0$ characterizing an expansion of a clade is defined as the ratio of $\beta$ and $\gamma$:

$$R_0 = \frac{\beta}{\gamma}. \tag{1}$$

However, for inference we re-parametrize the model in terms of the basic reproduction number $R_0$ and the net transmission rate $\tau$, similar to[31]. This allows us to build better interpretable priors, specifically for $R_0$. In addition, inference using the re-parametrized model is considerably faster due to the reduced complexity of priors required

for the identifiability of the model. The connection between these parameters and $\beta$ and $\gamma$ are presented in Equations (1) and (2).

$$\tau = \beta - \gamma \tag{2}$$

The general challenge in inferring parameters of a colonization model is that colonization is seldom under surveillance with sufficient genomic resolution. Consequently, an observation model is needed to connect the unobserved colonization cases to the number of BSI cases, which are assumed to be observed in systematic surveillance. Taken together, the colonization and observations models then allow us to quantify the number of BSI cases that would be expected under different combinations of $\beta$ and $\gamma$. We can thus infer the posterior distributions of these key parameters by identifying the $\beta$ and $\gamma$ such that the simulated BSI counts reproduce the observed BSI counts as accurately as possible. The model is fitted separately for the three ST131 clades of interest (A, C1 and C2) to compare their respective transmission efficiencies.

### Colonization model

The SIR-like model for the colonization by clade $c$ is defined by

$$
\begin{aligned}
dS_c/dt &= -\beta S_c C_c, \\
dC_c/dt &= \beta S_c C_c - \gamma C_c, \\
dR_c/dt &= \gamma C_c.
\end{aligned}
\tag{3}
$$

The $S_c$ compartment is the number of individuals at risk of being colonized by the clade $c$, while the $C_c$ compartment is the number of individuals colonized by the clade $c$. The structure of the colonization model is visualized on the first row of Fig. 1, where the horizontal arrows represent the flow of individuals from one compartment to the other. In practice, we discretize time and simulate transitions between compartments in steps of one week. This results in weekly counts of individuals in the different compartments, $S_c^t$, $C_c^t$, and $R_c^t$. Here, $t$ indexes the week, with $t = 1, \ldots, 728$ for ST131-A, corresponding to an observation period of 14 years ($\approx 52 \times 14$ weeks), from 2004 to 2017. For ST131-C2 the observation period is 11 years from 2007 to 2017, with $t = 1, \ldots, 572$. For ST131-C1, we continued the simulation for 5 years after the final observed BSI incidence at 2017, leading to a simulation period of 19 years with $t = 1, \ldots, 988$, which prevented outlier values appearing for the maximum BSI incidence during the observation time window (for details see the Inference subsection).

### Observation model

The observation model for BSI connects the simulated unobserved colonization events to observed BSI cases. The observations are modeled as Poisson distributed, where the mean $\bar{N}_{BSI, c}$ depends on the underlying colonization and a set of estimated parameters. Let $\rho_c$ denote the invasiveness of the clade of interest, $c$, relative to the other clades:

$$\rho_c = \frac{\bar{N}_{BSI, c}}{C_c} \bigg/ \frac{N_{BSI, 0}}{C_0}, \tag{4}$$

where $\bar{N}_{BSI, c}$ is the mean number of BSI cases caused by clade $c$, $N_{BSI, 0}$ is the number of BSI cases not associated with clade $c$, and, similarly, $C_c$, $C_0$ denote the numbers of colonization by clade $c$ or any other clade, respectively. In practice the quantities in (4) are defined for each week $t$, but the index $t$ is here suppressed for clarity. The relative invasiveness value $\rho_c$ is obtained from a previous study[16] (see below for details), and $N_{BSI, 0}$ is calculated as

$$N_{BSI, 0} = N\theta_{BSI} - \bar{N}_{BSI, c}, \tag{5}$$

where $N$ is the population size, and $\theta_{BSI}$ is the estimated proportion of the population with a BSI infection in a week. As an estimate of $\theta_{BSI}$ we use the weekly average number of BSI cases across the whole study period:

$$\theta_{BSI} = \frac{n}{wN}, \tag{6}$$

where $N$ is the population size, $n$ is the total number of BSI infections in $w$ weeks (728 for ST131-A, 988 for ST131-C1 and 572 for ST131-C2), obtained from the annual Norwegian national bacterial infection surveillance reports. Substituting Equations (5) and (6) into (4) and solving for $\bar{N}_{BSI, c}$ yields the mean of the Poisson model

$$\bar{N}_{BSI, c} = \frac{\rho_c C_c}{(C - C_c) + \rho_c C_c} N\theta_{BSI}, \tag{7}$$

which shows how the number of colonized individuals, $C_c$, is connected with the mean of BSI cases, $\bar{N}_{BSI, c}$. The observed BSI cases are thus assumed to be distributed as $N_{BSI, c} \sim \text{Pois}(\bar{N}_{BSI, c})$. Some intuition for this connection can be gained by considering the special case of clade $c$ being as invasive as all other clades, in which case $\rho_c = 1$, and (7) reduces to

$$\bar{N}_{BSI, c} = \frac{C_c}{C} N\theta_{BSI}. \tag{8}$$

In this special case the mean proportion of BSI infections caused by clade $c$ is simply equal to the proportion of colonization by clade $c$ among all *E. coli* colonizations. Equation (7) adjusts this proportion to take into account the invasiveness of the clade $c$, such the proportion of infections can be larger or smaller than what is implied by the proportion of colonization. In practise we use $C = C_0 + C_c$ and assume that $C = N$, i.e., everybody in the population is colonized by some *E. coli* which is justified by considerable amount of evidence in the literature[11].

The time index $t$ was excluded from the variables in the above definitions for simplicity, however, the observed data on BSI incidence are available as yearly aggregates. The discretized colonization model produces weekly colonization counts, so in order to compare the simulation and observations, we consequently aggregate the simulated weekly BSI counts by clade $c$, $N_{BSI, c}^t$ into a yearly count $R_{BSI, c}^{t'}$:

$$R_{BSI, c}^{t'} = \sum_{t \in T_{t'}} N_{BSI, c}^t, \tag{9}$$

where $T_{t'}$ is the set of all weeks in the $t'$-th year. As it is not plausible to assume that the population colonization process started simultaneously with the first BSI case occurrences, we introduce a discrete random variable $\Delta \sim \mathcal{U}\{0, 0.5 \cdot 52\}$ that indicates the number of weeks of colonization simulation before the observed data was gathered up to half a year earlier. Essentially the simulation is run for $t = -\Delta + 1, -\Delta + 2\ldots, 728$, where $t = 1$ is the first week of the first year of data collection $t' = 2004$. This summation is repeated for each year $t' \in \{2004, \ldots, 2017\}$ for ST131-A, $t' \in \{2004, \ldots, 2022\}$ for ST131-C1 and $t' \in \{2007, \ldots, 2017\}$ for ST131-C2.

### Data

A longitudinal genomic survey of BSI isolates in Norway collected between 2002–2017 without any phenotypic bias such as the presence of resistance to a particular antibiotic were used to estimate the incidence of ST131 clades in BSI per 1,000,000 individuals in Norway between years from 2004 to 2017[14]. The incidence/1,000,000 for total *E. coli* BSI was available from the Norwegian Surveillance of

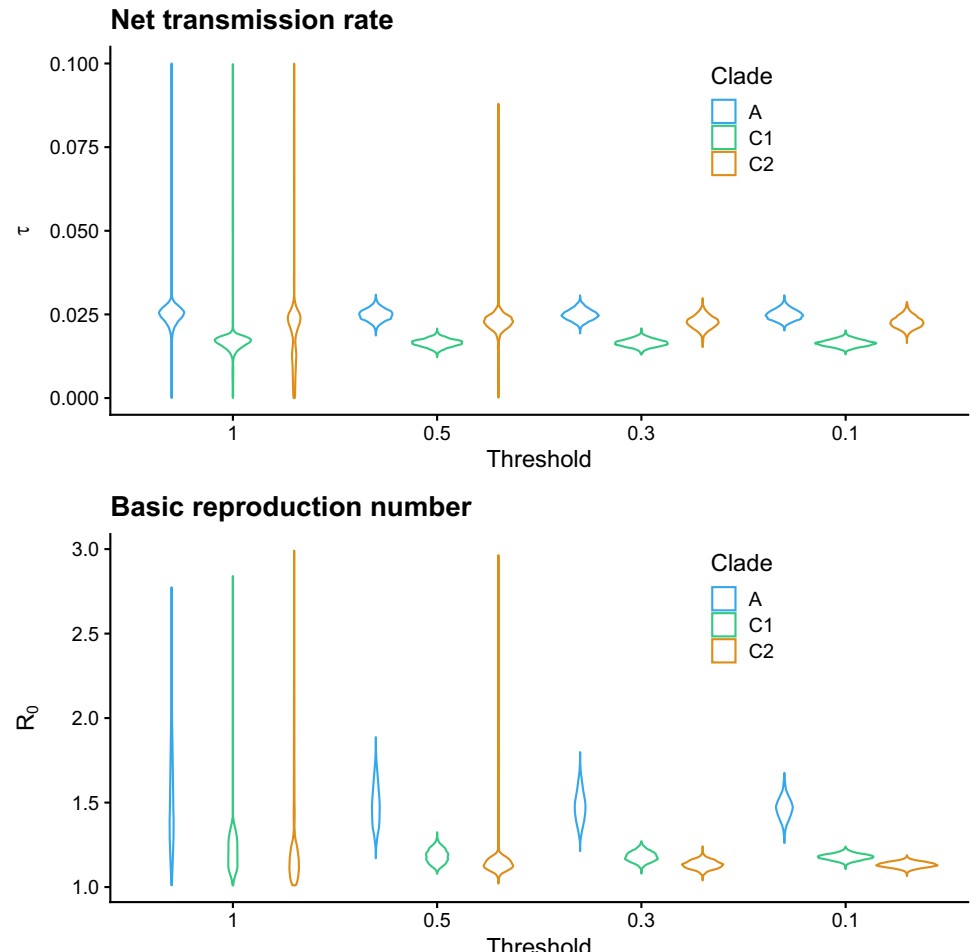

**Fig. 6 | Convergence of SMC ABC for net transmission rate $\tau$ and the basic reproduction number $R_0$.** Violin plots are shown for posterior approximations corresponding to the sampled four SMC particle populations with the thresholds 1, 0.5, 0.35 and 0.1 for ST131-A, ST131-C1 and ST131-C2.

antimicrobial resistance (NORM) reports[32]. This incidence is used to calculate the value of $\theta_{BSI}$ in (6). We previously estimated the relative invasiveness of different *E. coli* ST131 clades, specified by odds ratios that compare the frequency of each clade in geographically matched 2014-2017 neonatal gut colonization genomic data and longitudinal BSI 2003-2017 genomic data from the UK, after the emergence of ST131 and ST69 when a stable population equilibrium was reached[17,16]. These relative invasiveness values of different *E. coli* clades were used as estimates $\rho_c$ for the observation model. The odds ratio for any given clade is estimated by comparing its observed frequency in disease and healthy colonization using established methodology for opportunistic bacterial pathogens[33]. Further examination of the prevalence of most common STs in asymptomatic colonization in the UK neonates in comparison with healthy adult colonization in France using data collected in 2010[34] showed that the two countries have a highly similar clone distribution. This lends credibility to the assumption of the relative invasiveness estimates derived from the UK data being sufficiently representative in terms of geography and sampling age for use in modeling Norwegian BSI surveillance data.

### Inference

The likelihood function is intractable for the defined hybrid deterministic-stochastic model. However, since it can be implemented efficiently as a forward simulation, we use Approximate Bayesian Computation (ABC) likelihood-free inference to calibrate the model parameters in the light of data[19].

We parametrize the model in terms of the basic reproduction number $R_0$ and the net transmission rate $\tau$, similar to Lintusaari et al.[31]. For $R_0$, we use a truncated normal prior. The underlying distribution is $N(0, 1)$ restricted to [1.01, 3.0]. We set priors to the parameters of the model as follows:

$$\tau \sim \mathcal{U}(0, 0.1), \tag{10}$$

$$R_0 \sim \psi(0, 1, 1.01, 3) \tag{11}$$

In addition to these parameters that define the colonization model, we include the parameter $\Delta$ describes the delay between colonization and onset of infection. We assign $\Delta$ a uniform prior, allowing for a delay between 0 and 0.5 years. Given that the unit of $\Delta$ is one week, the upper limit of the uniform distribution equals 26 weeks.

$$\Delta \sim \mathcal{U}(0, 26) \tag{12}$$

Since $\Delta$ operates on a markedly different scale from $\tau$ and $R_0$, we scale it by 0.001 before fitting the model and then transform it back to the original scale for interpretation of the results.

To use ABC, following operational quantities are needed: 1) summary statistics to sufficiently reduce the dimension of the data, 2) a discrepancy function that allows comparing the simulated synthetic data summaries with the summaries of the observed data, and 3) a

threshold parameter that reflects tolerable error between the summarized synthetic and observed data, to accept the parameter that generated the simulation as a draw from the posterior. We constructed three summary statistics, $S_1$, $S_2$, $S_3$, to compare how the observed and simulated BSI incidences evolve in time. $S_1$ is defined as the maximum BSI incidence over the years within the interval [2004, 2017] for ST131-A, [2004, 2022] for ST131-C1 and [2007, 2017] for ST131-C2. $S_2$ is the number of years from the beginning of the simulation (year 2004 for ST131-A and 2007 for ST131-C2) to the maximum BSI incidence. $S_3$ is the BSI incidence at the beginning of the observation period. We apply $\log(x + 0.5)$-transformation on the summaries $S_1$ and $S_3$ because the scale of these BSI count -based summaries, which can vary between 0 and 1,000,000 (population size used in the simulation), is very different from the scale of the peaking time summary $S_2$ (ranging between 1 and 11, 14 or 19 years). Finally, the Euclidean distance was used to compare the observed and simulated data summaries.

Using sequential Monte Carlo (SMC) ABC[35], we obtained 10,000 samples as an approximation to the posterior distribution, with the following sequence of decreasing thresholds: [1.0, 0.5, 0.35, 0.1]. Convergence of the algorithm over these thresholds is visualized in Fig. 6. At the post-processing stage, the posteriors were resampled using the obtained weights. We used the ELFI implementation of the SMC ABC algorithm[36]. The model was fitted separately for clades A, C1 and C2 using a Python implementation of the simulator model and the ABC inference pipeline available in the open-source likelihood-free inference software package, Engine for Likelihood-free Inference (ELFI)[37]. The complete analysis pipeline written in Python is available from[38].

### Reporting summary

Further information on research design is available in the Nature Portfolio Reporting Summary linked to this article.

## Data availability

Source data are provided with this paper and are also available from (https://github.com/fanoja/Ecoli-R0[38]). Source data are provided with this paper.

## Code availability

Code for implementing, fitting and performing diagnostics on the model is available from (https://github.com/fanoja/Ecoli-R0[38]).

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

## Acknowledgements

Research in this publication was supported by the Research Council of Norway through its Centre of Excellence Integreat - The Norwegian Centre for knowledge-driven machine learning, project number 332645 (H.P., J.C.), Trond Mohn Foundation (BATTALION grant, J.C., R.A.G.) and Academy of Finland (EuroHPC grant T.M., J.C.; Flagship programme: Finnish Center for Artificial Intelligence FCAI (J.C., P.M.); and grants 358246, 352986 (F.O., P.M.)) and EU (H2020 grant 101016775 and NextGenerationEU, F.O., P.M.). F.O. acknowledges travel support from the European Union's Horizon 2020 research and innovation pro-gramme under ELISE Grant Agreement No 951847.

## Author contributions

F.O., H.P., P.M. and J.C. primarily developed the model and designed the inference procedure, F.O. and H.P. implemented and fitted the model, R.A.G. provided the epidemiological data and contributed to the model development and interpretation, G.T.H. and T.M. provided further expertise on colonization and transmission modeling, F.O., H.P., P.M. and J.C. wrote the manuscript which was further improved and revised by all named authors.

## Competing interests

The authors declare no competing interests.
