## [Peer Review file · Nature Communications]

Basic reproduction number varies markedly between closely related pandemic *Escherichia coli* clones

Corresponding Author: Professor Jukka Corander

Version 0:

Reviewer comments:

Reviewer #1

(Remarks to the Author)

Overall, this is an interesting and well written manuscript.

ST131-C1 is a very common and important global subclade. It would be important to include this subclade in the analysis. The results should mirror C2 and will act as "quality-assurance" for this model.

Successful antimicrobial susceptible *E. coli* clones (e.g., ST69, ST73 and ST95) but especially ST95 should be included the analysis. The results should mirror ST131-A and will act as "quality-assurance" for this model.

ST131-C1_M27 subclade and FQ-R ST1193 are currently emerging globally. Even though ST131-B is not a prominent clade, it is found among blood culture isolates (reference PMID: 33219650). It would be interesting to include them in the analysis.

Lines 100-1. ST131-C1 is as famous as ST131-C2.

Lines 103-12. That is a Scandinavian scenario; globally for the most part, ST131-C1 expanded in the early-mid 2000s, followed by ST131-C2 in the mid-late 2000s. ST131-A (and ST131-B), are rare in ST131 populations from humans outside Scandinavia.

(Remarks on code availability)

Outside the scope of my expertise.

Reviewer #2

(Remarks to the Author)

An interesting paper using a kind of "indirect observation" model to estimate properties of the colonization process in a population larger than the one exhibiting disease (BSI).

However, such inference will presumably be quite model dependent, wherefore some questions arise:

Comparisons of R_0 with influenza viruses could be complicated by the partial immunity to flu that parts of the population could have achieved by previous exposures or vaccinations. Does immunity have any role in *E. Coli* colonisation/clearance/disease/discovery for different clades ?

Since the process is extended over more than 10 years, isn't vital dynamics necessary?

The SIR model was originally intended for mostly airborne diseases. Doesn't the colonization process depend more on close contacts? Wouldn't, say, household infection be more relevant?

The introduction of a temporal delay in the model seems to indicate that it is the colonized individual who gets the disease. Is this reasonable in the case of nosocomial and similar spread ?

Fig. 3 observed data seems to suggest incidence of BSI peaking around 2013-15 and then declining. Why? This is related to comment on lines 160-163: if only the exponential phase was to be modelled, then why do Figs also show decline phase?

(Remarks on code availability)

Reviewer #3

(Remarks to the Author)

The analysis addresses an important and understudied aspect of bacterial transmission—specifically, the estimation of transmission in the context of the existing lack of systematic colonization data. This is a valuable contribution, as colonization dynamics are typically difficult to quantify and often overlooked. However, the modeling framework feels somewhat oversimplified given the biological complexity of clade interactions and the long timescale of the data. Additional analyses—such as sensitivity checks, alternative model structures, or incorporation of clade interactions—would strengthen the robustness of the conclusions. Overall, the study is an interesting step forward, but further work is needed to fully support its claims.

Major Comment:

The authors present a compartmental model, coupled with a stochastic observation process, to infer the transmission dynamics of *E. coli* clades ST131-A and ST131-C2 in Norway. While the approach is methodologically sound and addresses a compelling question, I have two major concerns that limit the strength of the conclusions.

1. Data limitations: The inference relies on prior estimates of *E. coli* colonization and invasiveness from the UK, as well as longitudinal records of bloodstream infections (BSI) from Norway. The authors should more explicitly discuss the potential mismatch in both age distribution and geographic context between these datasets. At a minimum, a sensitivity analysis should be conducted to assess the robustness of the results to assumptions about key parameters.

2. Model simplification: The compartmental model treats clades independently and does not incorporate potential interactions such as ecological competition, cross-immunity, or within-host dynamics. These omissions may overlook key biological processes that could explain patterns such as the apparent replacement of ST131-A by ST131-C2. Incorporating such interactions, or at least discussing their potential impact, would significantly strengthen the interpretation of the model's results.

Despite these concerns, the study addresses an important question and is worthy of publication, provided these limitations are more thoroughly addressed through additional analysis and discussion.

Comments on individual sections:

Results

Although the model is reparametrized in terms of the basic reproduction number (R_0) and the net transmission rate (τ), the authors should also report estimates for the duration of colonization—specifically, the posterior distribution of the recovery rate $1/\gamma$ (often denoted as D) and of the parameter Δ describes the delay between colonization and onset of infection. These parameters are biologically meaningful and directly interpretable, and reporting them would allow for comparison with published estimates of *E. coli* colonization duration in different populations. Including this discussion would help contextualize the model assumptions and better assess their plausibility in light of empirical evidence.

3.2 Colonization Model

The authors adopt a relatively simple modeling structure, using clade-specific SIR models to capture the transmission dynamics of individual *E. coli* clades. Although they acknowledge that reinfection is possible, they justify the use of an SIR framework by arguing that recolonization is negligible over the time frame of the simulation. However, given that the observation window spans up to 13 years, it is not clear that this can reasonably be considered a "short" period in which reinfection can be safely ignored. Is there empirical evidence suggesting that recolonization is truly a rare event over such a timespan? Without supporting data, this assumption may oversimplify the natural history of colonization and transmission. It appears that the colonization and observation models are fitted independently for each clade. While I understand that incorporating interaction terms—such as competition, cross-immunity, or within-host competition—would significantly increase model complexity, omitting these interactions may overlook important dynamics. In particular, if reinfection is considered negligible (as the authors argue to justify using an SIR model), it raises the question of why cross-protection between clades is also assumed to be negligible. Is susceptibility to clade i truly independent of prior or current colonization with clade j ? The authors claim that the short observation window makes it reasonable to ignore re-colonization, but does this same timeframe also justify assuming no cross-protection or competition? The observed delayed growth patterns—where ST131-A initially dominates but then declines as ST131-C2 expands—might be better explained by clade competition. Ignoring interactions may therefore obscure key epidemiological mechanisms shaping the dynamics of these lineages.

3.3 Observational model

The manuscript relies on a combination of longitudinal BSI data from Norway and previously published estimates of clade-specific invasiveness derived from UK datasets to inform the observation model through the parameter ρ_0 . While this is a reasonable approach given the limited availability of colonization data, it introduces several important limitations that warrant more explicit discussion. The estimate of ρ_0 is based on comparisons between neonatal gut colonization and bloodstream

infection (BSI) data from the UK, where BSI cases predominantly occur in older adults. This creates a potential mismatch both in age distribution and geographic context. The authors should discuss the assumption that clade-specific invasiveness is uniform across age groups, especially given known differences in immune status and comorbidity burden across the life course. For example, is it plausible that certain clades are more invasive in elderly individuals due to immunosenescence or underlying conditions?

Second, geographic variation in healthcare systems, antimicrobial use, and background strain prevalence may affect both colonization dynamics and the probability of progression to invasive disease. Estimates derived from the UK may therefore not fully generalize to the Norwegian setting. While the model assumes that relative invasiveness (ρ_0) is a stable biological feature of each clade, ecological and healthcare-related factors might influence these probabilities in practice. Third, the use of a stable population equilibrium as the basis for estimating ρ_0 introduces further assumptions. It presumes that transmission and selection pressures remained relatively constant during the observation window and that the bacterial population had stabilized—an assumption that may not be strictly valid, especially in the context of recent clonal expansions or changes in clinical practice.

3.4 Data

Section 3.4 would benefit from expansion to ensure the manuscript is fully self-contained. Although the relative invasiveness parameter (ρ_0) is derived from a previous study, the current manuscript should provide a more detailed summary of how this estimate was obtained. Specifically, it would be helpful to clarify the population studied, the data sources (e.g., neonatal colonization vs. BSI surveillance), and the assumptions underlying the derivation of ρ_0 . Additionally, any known limitations of the dataset—particularly in relation to population representativeness, age structure, or geographic context—should be explicitly acknowledged either in this section or more thoroughly in the Discussion. This would improve transparency and help readers assess the robustness of the model's conclusions.

(Remarks on code availability)

Version 1:

Reviewer comments:

Reviewer #3

(Remarks to the Author)

The authors have adequately addressed my comments and concerns.

(Remarks on code availability)

Reviewer #4

(Remarks to the Author)

I agree with the authors for no opportunity to include ST69, ST73 and ST95 for comparison due to the lack of continued expansion signals. Considering the vague expansion signal, the exclusion of ST131-B is also reasonable. Nevertheless, these highlight a limitation of the proposed model. I suggest that the authors add a section in Discussion to address the limitations of their models.

(Remarks on code availability)

Authors' Response to Reviews of

Basic reproduction number for pandemic *Escherichia coli* clones varies markedly and can be comparable to pandemic influenza viruses

Fanni Ojala, Henri Pesonen, Rebecca A. Gladstone, Tommi Mäklin, Gerry Tonkin-Hill, Pekka Marttinen, Jukka Corander
Nature Communications,

RC: *Reviewers' Comment*, AR: Authors' Response, □ Manuscript Text

NB larger added text sections are marked in red in the revised manuscript, more minor edits are unmarked.

1. Reviewer 1

RC: *Overall, this is an interesting and well written manuscript.*

AR: We thank you for the kind feedback.

RC: *ST131-C1 is a very common and important global subclade. It would be important to include this subclade in the analysis. The results should mirror C2 and will act as "quality-assurance" for this model.*

AR: We have now estimated R_0 using our modeling framework for ST131-C1. Indeed, the estimated R_0 values of ST131-C1 are close to that of clade ST131-C2.

RC: *Successful antimicrobial susceptible E. coli clones (e.g., ST69, ST73 and ST95) but especially ST95 should be included the analysis. The results should mirror ST131-A and will act as "quality-assurance" for this model.*

AR: We agree that those clones would be interesting to consider, however, the dating analysis in the recently published article (PMID 40180894) shows that ST73 and ST95 expanded a long time ago, and when we examined ST69 BSI counts in the Norwegian cohort also this clone showed no longer any signal of continued expansion, unlike the subclades of ST131. Hence, there is unfortunately no opportunity to include these other clones for comparison.

RC: *ST131-C1_M27 subclade and FQ-R ST1193 are currently emerging globally. Even though ST131-B is not a prominent clade, it is found among blood culture isolates (reference PMID: 33219650). It would be interesting to include them in the analysis.*

AR: We were unable to include clade ST131-B, since the data does not contain a clear epidemic peak for that clade. As estimated in Gladstone et al. using a coalescent-based model (PMID 35544167), ST131-B expanded roughly a decade earlier compared with its sister subclades, leaving the expansion signal too vague for fitting an epidemiological model.

RC: *Lines 100-1. ST131-C1 is as famous as ST131-C2.*

AR: We have modified the text in accordance to this comment.

RC: *Lines 103-12. That is a Scandinavian scenario; globally for the most part, ST131-C1 expanded in the early-mid 2000s, followed by ST131-C2 in the mid-late 2000s. ST131-A (and ST131-B), are rare in ST131 populations from humans outside Scandinavia.*

AR: We have edited the text to avoid misleading impression about the geographical context considered. Note, however, that based on Mäklin et al. (PMID 36456554) ST131-A is by far the most common coloniser of humans in the UK out of the four clades of ST131 .

2. Reviewer 2

RC: *An interesting paper using a kind of "indirect observation" model to estimate properties of the colonization process in a population larger than the one exhibiting disease (BSI). However, such inference will presumably be quite model dependent, wherefore some questions arise: Comparisons of R_0 with influenza viruses could be complicated by the partial immunity to flu that parts of the population could have achieved by previous exposures or vaccinations. Does immunity have any role in E. Coli colonisation/clearance/disease/discovery for different clades ?*

AR: It is correct that direct comparisons of our results with for example flu viruses are challenging so we have added a statement about caution being needed in interpretation in this regard into the Discussion. As with all infectious diseases, R_0 is a model-based quantity derived from a set of assumptions that are never exactly met in reality. In our opinion, the most important use of R_0 estimates is to provide a relative ranking of the transmission capabilities for a set of microbes, when everything else is held constant. In the revision we demonstrate that the interpretation of A clade being significantly more transmissible than the other two clades (C1, C2) is robust with respect to a number of key assumptions used in the modeling. We have edited the Abstract and Discussion to highlight this and to avoid over-interpretation of the exact values of R_0 obtained from the model.

RC: *Since the process is extended over more than 10 years, isn't vital dynamics necessary?*

AR: This is a relevant comment, however, we do think that our findings concerning clade-wise differences in relative transmissibility are not impacted by not including vital dynamics explicitly in the model.

RC: *The SIR model was originally intended for mostly airborne diseases. Doesn't the colonization process depend more on close contacts? Wouldn't, say, household infection be more relevant?*

AR: It is true that SIR model has mostly been used for airborne diseases, but it has also been used for estimating R_0 in settings where contact with body fluids is needed for transmission. Any contacts where fecal-oral route of transmission is feasible are relevant for ExPEC, however, it is indeed true that households are considered as transmission hotspots for these bacteria so we have amended the text accordingly.

RC: *The introduction of a temporal delay in the model seems to indicate that it is the colonized individual who gets the disease. Is this reasonable in the case of nosocomial and similar spread ?*

AR: Colonization does not necessarily lead to BSI, but we assume it precedes it, as this is typical for all opportunistic pathogenic bacteria. As shown, incidence of colonization is considerably higher than incidence of BSI.

RC: *Fig. 3 observed data seems to suggest incidence of BSI peaking around 2013-15 and then declining. Why?*

This is related to comment on lines 160-163: if only the exponential phase was to be modelled, then why do Figs also show decline phase?

AR: The decline is only seen for the BSI cases caused by ST131-A (Fig 4 in the revision). However, since there are data from only a couple of years post-expansion, sampling variation could explain this pattern. Continued genomic surveillance of disease cases beyond 2017 could help to resolve whether there has been a genuine decline or whether the clades have continued to circulate in the population at the same frequency beyond peaking. Note that the SIR compartmental model structure includes a peak in infections (in our case, colonization), therefore there is a declining phase in the predicted colonization and BSI beyond the peak. We have now clarified this point in the main text.

3. Reviewer 3

RC: *The analysis addresses an important and understudied aspect of bacterial transmission—specifically, the estimation of transmission in the context of the existing lack of systematic colonization data. This is a valuable contribution, as colonization dynamics are typically difficult to quantify and often overlooked. However, the modeling framework feels somewhat oversimplified given the biological complexity of clade interactions and the long timescale of the data. Additional analyses—such as sensitivity checks, alternative model structures, or incorporation of clade interactions—would strengthen the robustness of the conclusions. Overall, the study is an interesting step forward, but further work is needed to fully support its claims.*

3.1. Major comments

RC: *The authors present a compartmental model, coupled with a stochastic observation process, to infer the transmission dynamics of *E. coli* clades ST131-A and ST131-C2 in Norway. While the approach is methodologically sound and addresses a compelling question, I have two major concerns that limit the strength of the conclusions.*

RC: *1. Data limitations: The inference relies on prior estimates of *E. coli* colonization and invasiveness from the UK, as well as longitudinal records of bloodstream infections (BSI) from Norway. The authors should more explicitly discuss the potential mismatch in both age distribution and geographic context between these datasets. At a minimum, a sensitivity analysis should be conducted to assess the robustness of the results to assumptions about key parameters.*

AR: Thank you for highlighting these important steps that are necessary for ensuring robustness of our main conclusions. We have conducted a sensitivity analysis to make sure the results are not confounded by specific choices of values and priors for the key parameters (presented as Supplementary Information for the revision). Also, while preparing a revision of Gladstone et al. 2024 (<https://www.medrxiv.org/content/10.1101/2024.11.22.24317484v1>) we compared *E. coli* ST frequency data in asymptomatic colonization between the UK and France using published data (PMID 35862685, data from year 2010 included), which showed that the 10 most common STs had very similar population frequencies in the two countries, supporting our assumption of the relative invasiveness of clones being sufficiently robustly quantified for use in modelling the Norwegian data. Further, since the French colonization data are obtained from adults it lends credibility to the assumption that the neonatal colonization is reflective of adult colonization since the infants would most typically get colonized by the *E. coli* colonizing their parents and other family members.

RC: *Model simplification: The compartmental model treats clades independently and does not incorporate*

potential interactions such as ecological competition, cross-immunity, or within-host dynamics. These omissions may overlook key biological processes that could explain patterns such as the apparent replacement of ST131-A by ST131-C2. Incorporating such interactions, or at least discussing their potential impact, would significantly strengthen the interpretation of the model's results. Despite these concerns, the study addresses an important question and is worthy of publication, provided these limitations are more thoroughly addressed through additional analysis and discussion.

AR: Including potential interactions to the model would complicate the model structure considerably. We believe that the current version of the model serves as a proof-of-concept and can be improved on in the future. As already highlighted in the response to comments from other reviewers, we have amended the text to emphasize caution in the interpretation of the estimated R_0 values. We have now further added a discussion of the possibility to consider more complex models in the future that would expand on the simple compartmental structure. Note also that there is no evidence of ST131-A being replaced by the other clades based on the analysis of UK colonization data by Mäklin et al. (PMID 36456554).

3.2. Comments on individual sections

RC: *Although the model is reparametrized in terms of the basic reproduction number (R) and the net transmission rate (β), the authors should also report estimates for the duration of colonization—specifically, the posterior distribution of the recovery rate $1/D$ (often denoted as D) and of the parameter τ describes the delay between colonization and onset of infection. These parameters are biologically meaningful and directly interpretable, and reporting them would allow for comparison with published estimates of *E. coli* colonization duration in different populations. Including this discussion would help contextualize the model assumptions and better assess their plausibility in light of empirical evidence.*

AR: We have now included the posterior distribution of the duration of colonization in our results. Our results indicated that the delay parameter is not well-identifiable unlike the other key parameters, however, the sensitivity analysis further demonstrated that the main results remain robust to varying prior distribution for the duration of colonization.

3.2.1 3.2 Colonization Model

RC: *The authors adopt a relatively simple modeling structure, using clade-specific SIR models to capture the transmission dynamics of individual *E. coli* clades. Although they acknowledge that reinfection is possible, they justify the use of an SIR framework by arguing that recolonization is negligible over the time frame of the simulation. However, given that the observation window spans up to 13 years, it is not clear that this can reasonably be considered a "short" period in which reinfection can be safely ignored. Is there empirical evidence suggesting that recolonization is truly a rare event over such a timespan? Without supporting data, this assumption may oversimplify the natural history of colonization and transmission.*

AR: We have now included a comparison with the SIS-based model for the colonization part and added the results in the Supplementary Information. Our results indicated a substantially poorer fit for the alternative model.

RC: *It appears that the colonization and observation models are fitted independently for each clade. While I understand that incorporating interaction terms—such as competition, cross-immunity, or within-host competition—would significantly increase model complexity, omitting these interactions may overlook*

important dynamics. In particular, if reinfection is considered negligible (as the authors argue to justify using an SIR model), it raises the question of why cross-protection between clades is also assumed to be negligible. Is susceptibility to clade i truly independent of prior or current colonization with clade j ? The authors claim that the short observation window makes it reasonable to ignore re-colonization, but does this same timeframe also justify assuming no cross-protection or competition? The observed delayed growth patterns—where ST131-A initially dominates but then declines as ST131-C2 expands—might be better explained by clade competition. Ignoring interactions may therefore obscure key epidemiological mechanisms shaping the dynamics of these lineages.

AR: As written above, we have now included a SIS model as an alternative for the colonization compartment to allow for the effect of re-colonization to be taken into account, however, the SIS-based model showed much poorer fit. Since inclusion of clade interactions would considerably complicate the modeling task, we have instead added this as a possibility for future research into Discussion, as suggested by the reviewer.

3.2.2 3.3 Observation model

RC: *The manuscript relies on a combination of longitudinal BSI data from Norway and previously published estimates of clade-specific invasiveness derived from UK datasets to inform the observation model through the parameter ρ_c . While this is a reasonable approach given the limited availability of colonization data, it introduces several important limitations that warrant more explicit discussion. The estimate of ρ_c is based on comparisons between neonatal gut colonization and bloodstream infection (BSI) data from the UK, where BSI cases predominantly occur in older adults. This creates a potential mismatch both in age distribution and geographic context. The authors should discuss the assumption that clade-specific invasiveness is uniform across age groups, especially given known differences in immune status and comorbidity burden across the life course. For example, is it plausible that certain clades are more invasive in elderly individuals due to immunosenescence or underlying conditions?*

AR: These are all warranted concerns and we have taken steps to ensure robustness of our assumptions with respect to them (see the response above to the first comment on Data limitations).

RC: *Second, geographic variation in healthcare systems, antimicrobial use, and background strain prevalence may affect both colonization dynamics and the probability of progression to invasive disease. Estimates derived from the UK may therefore not fully generalize to the Norwegian setting. While the model assumes that relative invasiveness (ρ_c) is a stable biological feature of each clade, ecological and healthcare-related factors might influence these probabilities in practice.*

AR: These are all warranted concerns and we have taken steps to ensure robustness of our assumptions with respect to them (see the response above to the first comment on Data limitations). However, we have also added a comment in the Discussion on the potential influence of these differences on the results.

RC: *Third, the use of a stable population equilibrium as the basis for estimating ρ_c introduces further assumptions. It presumes that transmission and selection pressures remained relatively constant during the observation window and that the bacterial population had stabilized—an assumption that may not be strictly valid, especially in the context of recent clonal expansions or changes in clinical practice.*

AR: This is true and we have added a comment about it in the Discussion. Note however that we took steps to avoid biases emerging from clonal expansions in the original work estimating the odds ratios by including only infection data from the post clade-expansion period in the UK (colonization samples were collected after the clade prevalences had stabilized).

3.2.3 3.4 Data

RC: *Section 3.4 would benefit from expansion to ensure the manuscript is fully self-contained. Although the relative invasiveness parameter (ρ_c) is derived from a previous study, the current manuscript should provide a more detailed summary of how this estimate was obtained. Specifically, it would be helpful to clarify the population studied, the data sources (e.g., neonatal colonization vs. BSI surveillance), and the assumptions underlying the derivation of ρ_c .*

AR: We have added more details on this in the revision.

RC: *Additionally, any known limitations of the dataset—particularly in relation to population representativeness, age structure, or geographic context—should be explicitly acknowledged either in this section or more thoroughly in the Discussion. This would improve transparency and help readers assess the robustness of the model's conclusions.*

AR: See above responses.

References

Response to reviewer comments

Reviewer #3 (Remarks to the Author):

The authors have adequately addressed my comments and concerns.

Reviewer #4 (Remarks to the Author):

I agree with the authors for no opportunity to include ST69, ST73 and ST95 for comparison due to the lack of continued expansion signals. Considering the vague expansion signal, the exclusion of ST131-B is also reasonable. Nevertheless, these highlight a limitation of the proposed model. I suggest that the authors add a section in Discussion to address the limitations of their models.

We agree with the reviewer and have accordingly added text in the Discussion to address this limitation of the introduced model in comparison to phylogenetic models.